# TAPBPR alters MHC class I peptide presentation by functioning as a peptide exchange catalyst

Clemens Hermann[1†‡], Andy van Hateren[2†], Nico Trautwein[3], Andreas Neerincx[1], Patrick J Duriez[4], Stefan Stevanović[3], John Trowsdale[1], Janet E Deane[5], Tim Elliott[2], Louise H Boyle[1*]

[1]Department of Pathology, University of Cambridge, Cambridge, United Kingdom; [2]Faculty of Medicine and Institute for Life Science, University of Southampton, Southampton, United Kingdom; [3]Department of Immunology, Eberhard Karls University Tübingen, Tübingen, Germany; [4]Cancer Research UK Protein Core Facility, Faculty of Medicine, University of Southampton, Southampton, United Kingdom; [5]Cambridge Institute for Medical Research, University of Cambridge, Cambridge, United Kingdom

*For correspondence: lhb22@cam.ac.uk

†These authors contributed equally to this work

Present address: ‡Department of Integrative Biomedical Sciences, Division of Chemical and Systems Biology, Institute for Infectious Disease and Molecular Medicine, University of Cape Town, Cape Town, South Africa

Competing interests: The authors declare that no competing interests exist.

**Abstract** Our understanding of the antigen presentation pathway has recently been enhanced with the identification that the tapasin-related protein TAPBPR is a second major histocompatibility complex (MHC) class I-specific chaperone. We sought to determine whether, like tapasin, TAPBPR can also influence MHC class I peptide selection by functioning as a peptide exchange catalyst. We show that TAPBPR can catalyse the dissociation of peptides from peptide-MHC I complexes, enhance the loading of peptide-receptive MHC I molecules, and discriminate between peptides based on affinity in vitro. In cells, the depletion of TAPBPR increased the diversity of peptides presented on MHC I molecules, suggesting that TAPBPR is involved in restricting peptide presentation. Our results suggest TAPBPR binds to MHC I in a peptide-receptive state and, like tapasin, works to enhance peptide optimisation. It is now clear there are two MHC class I specific peptide editors, tapasin and TAPBPR, intimately involved in controlling peptide presentation to the immune system.

## Introduction

Major histocompatibility complex (MHC) class I molecules convey a selection of the peptidome of a cell to the immune system. As well as being of crucial importance for the detection of intracellular pathogens, particularly viruses, it is now apparent that peptide presentation by MHC class I is extremely relevant in the recognition of tumours (*Rötzschke et al., 1990*; *Duan et al., 2014*; *Gubin et al., 2014*; *Snyder et al., 2014*; *Yadav et al., 2014*). Despite this, the molecular mechanisms controlling peptide selection for immune recognition are still poorly understood. Currently, the loading of peptide onto MHC class I is known to be orchestrated by the co-factor tapasin and is thought to occur predominantly within the peptide loading complex (PLC) in the endoplasmic reticulum (*Sadasivan et al., 1996*; *Li et al., 1997*; *Ortmann et al., 1997*; *Lehner et al., 1998*; *Tan et al., 2002*). Tapasin enhances the rate and the extent of MHC class I peptide loading and improves the discrimination that occurs between peptides to ensure MHC class I molecules are loaded with high-affinity peptides, thus prolonging cell surface expression of MHC class I molecules (*Williams et al., 2002*; *Howarth et al., 2004*; *Chen and Bouvier, 2007*; *Wearsch and Cresswell, 2007*; *van Hateren et al., 2013*). MHC class I allomorphs differ in their dependence on tapasin for efficient

**eLife digest** Our immune system protects us from infections and destroys cells that are turning cancerous. A group of proteins called MHC class I molecules are essential for this protection. These molecules let the immune system know what is going on inside our cells by displaying chopped up fragments of proteins (or peptides) on the surface of the cell. If these peptides are from infectious or disease-causing agents the immune system is triggered into action and can recognise and kill the cell.

There is still much to discover regarding how MHC molecules choose which peptides to display. A very complex pathway within our cells controls this displaying of peptides to the immune system. Recently, a protein called TAPBPR was identified as a new player in MHC class I biology, but its role was unclear.

Hermann, van Hateren et al. now reveal that TAPBPR plays a central role in restricting which peptides are loaded onto and presented by MHC class I molecules. The results suggest that TAPBPR acts as a quality control checkpoint, closely monitoring and ensuring that the loaded peptide is stable within the MHC class I molecule. The discovery of TAPBPR's role in peptide selection increases our understanding of how peptides are chosen and stabilized, and sets the stage for learning more about how cells decide which peptides to reveal to the immune system.

peptide loading (*Greenwood et al., 1994*; *Peh et al., 1998*; *Williams et al., 2002*). Furthermore, the ability to select and assemble with an optimal peptide cargo in the absence of tapasin is inversely correlated to the enhancement observed in the presence of tapasin (*Williams et al., 2002*; *van Hateren et al., 2013*; *Rizvi et al., 2014*).

It has recently become apparent that, in addition to tapasin, there is a second MHC class I-specific chaperone, the tapasin-related protein TAPBPR (*Boyle et al., 2013*). In contrast to tapasin, TAPBPR is not an integral component of the peptide loading complex and cannot compensate for the loss of tapasin (*Boyle et al., 2013*; *Hermann et al., 2013*). Currently, the precise function of TAPBPR in the MHC class I pathway is unknown (*Hermann et al., 2015*). A functional role in MHC class I antigen presentation is supported by the finding that the presence of TAPBPR slows the anterograde trafficking of MHC class I and prolongs the association of MHC class I with the PLC. This raises the possibility that TAPBPR serves as an additional quality control checkpoint in the MHC class I antigen presentation pathway (*Boyle et al., 2013*). Although tapasin and TAPBPR only share 22% identity (*Teng et al., 2002*), both bind MHC class I, and their orientation on MHC class I is similar (*Dong et al., 2009*; *Hermann et al., 2013*). This shared orientation raises the possibility that there is some common functionality between the two molecules, particularly in regard to the ability to influence the peptide repertoire presented on MHC class I. Here, we sought to determine whether, like tapasin, TAPBPR has a peptide editing function.

## Results

### Expression and purification of TAPBPR

In order to test whether TAPBPR has peptide editing functionality we chose an approach analogous to that previously established for tapasin, namely a cell free in vitro assay that allows monitoring of peptide binding to MHC class I in real time via fluorescence anisotropy (*Chen and Bouvier, 2007*). Due to the intrinsically low affinity between tapasin and MHC class I molecules in vitro, soluble tapasin function on MHC class I is only measurable when the two molecules are brought into close proximity via a Jun/Fos leucine zipper (*Chen and Bouvier, 2007*) or when tapasin is conjugated to ERp57 (*Wearsch and Cresswell, 2007*). Since TAPBPR is not integrated into the PLC and can be co-immunoprecipitated with MHC I (*Boyle et al., 2013*, *Hermann et al., 2013*), it is likely that TAPBPR has a higher affinity for MHC class I than tapasin. We therefore tested whether TAPBPR functioned in vitro without the need for an artificial intermolecular tether.

To this end, the luminal domains of TAPBPR were cloned into the pHLsec expression vector and transiently transfected into HEK293F cells. This resulted in the efficient production of a secreted

form of TAPBPR with a C-terminal His-tag, allowing for purification from the culture supernatant using Ni-affinity and size exclusion chromatography. TAPBPR protein eluted as a single major peak of high purity as verified by Coomassie staining following sodium dodecyl sulfate polyacrylamide gel electrophoresis (SDS-PAGE) (*Figure 1A*). TAPBPR[TN5] in which the isoleucine at position 261 was mutated to lysine, which inhibits the interaction of TAPBPR with MHC class I (*Hermann et al., 2013*), was produced in the same manner and was also efficiently expressed and produced a major single protein peak following size-exclusion chromatography (*Figure 1B*). Differential scanning fluorimetry revealed melting temperatures for TAPBPR and TAPBPR[TN5] to be between 51.5°C and 52.5°C indicating that the single point mutation did not adversely affect TAPBPR folding (*Figure 1C*). These mammalian-expressed, purified TAPBPR proteins were used in subsequent fluorescence anisotropy experiments to characterise the equilibrium and kinetic parameters between MHC class I and different peptide ligands.

## TAPBPR enhances peptide dissociation from HLA-A2 in vitro

To investigate if TAPBPR is able to directly edit peptides on MHC class I molecules in a manner similar to that attributed to tapasin, we adapted the in vitro peptide exchange assay previously developed for measuring tapasin function (*Chen and Bouvier, 2007*). Using fluorescent polarisation we tested whether TAPBPR enhanced peptide dissociation from HLA-A2, an allomorph that co-immunoprecipitates readily with TAPBPR (*Hermann et al., 2013*). HLA-A*02:01 molecules were loaded with the fluorescent peptide FLPSDC*FPSV (C* indicates the position of the fluorophore) before excess unlabelled FLPSDCFPSV peptide was added in the presence or absence of TAPBPR. Dissociation of FLPSDC*FPSV was only apparent in the first six hours in the presence of competitor peptide and TAPBPR (*Figure 2A*). Dissociation of FLPSDC*FPSV was also observed when a vast excess of unlabelled NLVPMVATV was used in the presence of TAPBPR (*Figure 2B*). To ensure the enhancement of peptide dissociation was a consequence of direct interaction between TAPBPR and HLA-A2, we repeated the peptide dissociation assays using TAPBPR[TN5] (I261K) which inhibits the ability of TAPBPR to bind to HLA-A2 (*Hermann et al., 2013*). No enhancement in the dissociation of FLPSDC*FPSV was observed in the presence of TAPBPR[TN5] (red line *Figure 2B*). Thus the ability of TAPBPR to enhance peptide dissociation is a direct consequence of TAPBPR binding to HLA-A2.

## TAPBPR enhances peptide association with HLA-A2 in vitro

Next we examined the effect that TAPBPR has on the kinetics of peptide association with HLA-A2. To test this, HLA-A*02:01 molecules refolded with a UV conditional ligand were made peptide-receptive by photolysis and the binding of the fluorescent peptide FLPSDC*FPSV was monitored by fluorescence polarisation in the absence or presence of TAPBPR. We found that TAPBPR catalysed binding of FLPSDC*FPSV to peptide-receptive HLA-A*02:01 (*Figure 2C*). In contrast, no enhancement in the association of FLPSDC*FPSV with HLA-A2 was observed in the presence of TAPBPR[TN5]

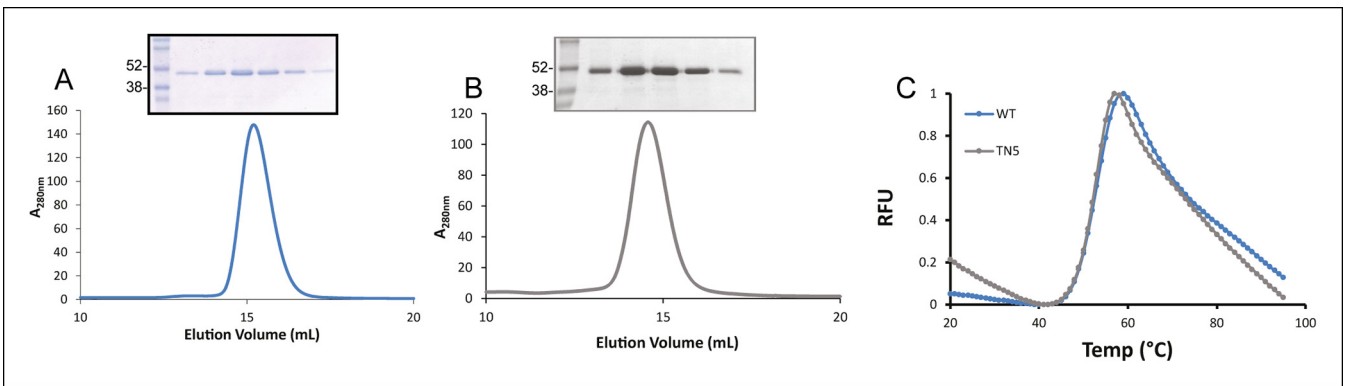

**Figure 1.** Expression of TAPBPR. Size exclusion chromatograms of (A) TAPBPR and (B) TAPBPR[TN5] purified from cell culture supernatants. The protein peaks were analysed by SDS-PAGE followed by Coomassie staining. (C) Differential scanning fluorimetry of TAPBPR and TAPBPR[TN5] demonstrate equivalent thermal denaturation profiles. The data is representative of two independent experiments.

(*Figure 2C*), further demonstrating that a direct association between TAPBPR and HLA-A2 is required for TAPBPR to influence HLA-A2 peptide binding characteristics.

## TAPBPR enhances the selection of high affinity peptide by HLA-A2 in vitro

To ascertain whether TAPBPR is able to discriminate between peptides, competition assays were performed in which peptide-receptive HLA-A*02:01 molecules were incubated with a mix of two peptides; 0.125 µM of the high affinity labelled peptide FLPSDC*FPSV, and a variable concentration of the peptide NLVPMVATV, which binds HLA-A*02:01 with lower affinity than FLPSDCFPSV. In the presence of TAPBPR, NLVPMVATV became a poorer competitor (*Figure 2D*). TAPBPR$^{TN5}$ had no

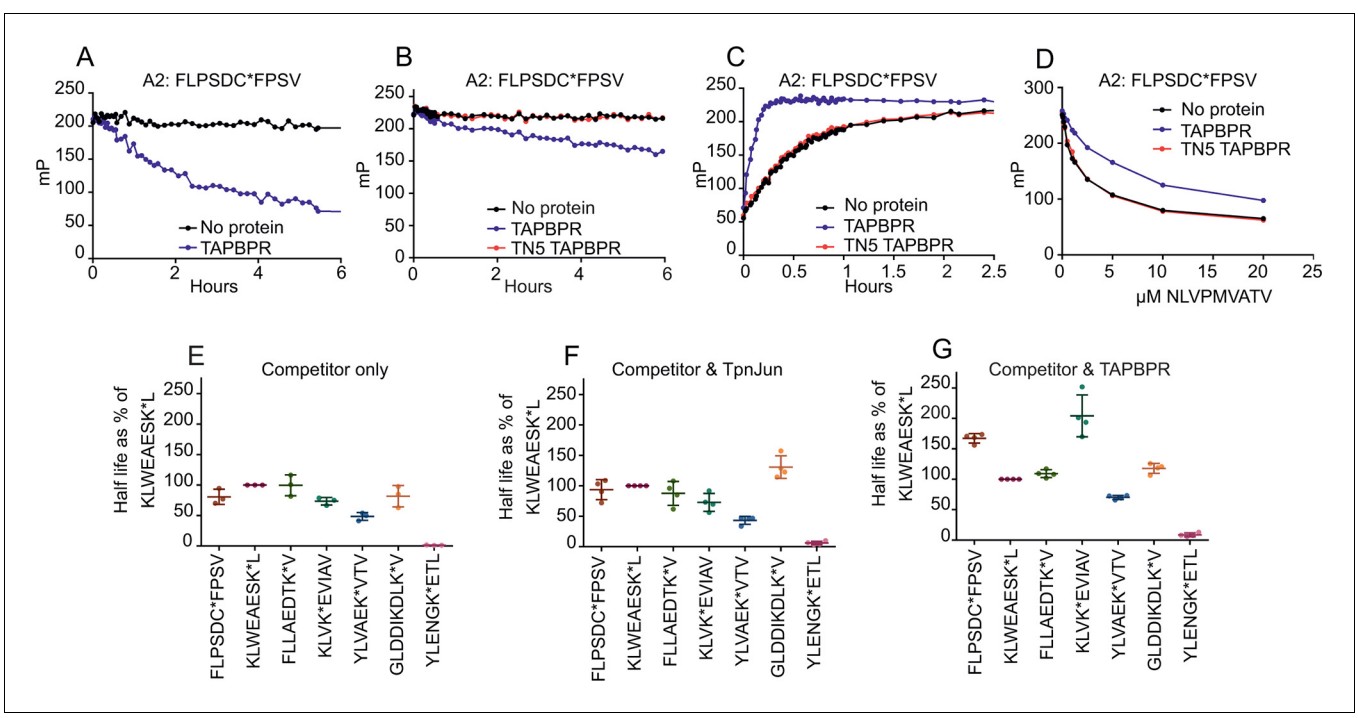

**Figure 2.** TAPBPR functions as a peptide loading catalyst and peptide editor for HLA-A2. (A,B) Dissociation, (C) association and (D) peptide competition of fluorescent peptide FLPSDC*FPSV on HLA-A*02:01 in the absence or presence of TAPBPR or TAPBPR$^{TN5}$ as measured by fluorescence polarisation. (A) 0.15 µM HLA-A*02:01fos molecules were mixed with 1.2 µM human β2m and loaded with 0.1 µM FLPSDC*FPSV and (B) 0.5 µM HLA-A*02:01 molecules were loaded with 0.125 µM FLPSDC*FPSV. The complexes were then split and incubated with 1000 fold molar excess of (A) FLPSDCFPSV or (B) NLVPMVATV with either A) buffer (No protein) or supplemented with 0.75 µM TAPBPR, or (B) buffer (No protein) or supplemented with 0.25 µM TAPBPR or TAPBPR$^{TN5}$. Note, the slight difference in dissociation rate observed in A & B is likely to be related to the concentration of TAPBPR used in each experiment and not to the sequence of the competing peptide used. The data shown in *Figure 2A,B* is representative of 13 independent experiments, which all produced similar results. (C) 0.5 µM HLA-A*02:01 molecules were made peptide receptive and then the binding of 0.125 µM FLPSDC*FPSV was followed in the presence or absence of 0.05 µM TAPBPR or TAPBPR$^{TN5}$. One representative association experiment of 13 experiments is shown. (D) 0.5 µM HLA-A*02:01 molecules were made peptide-receptive and incubated with 0.125 µM high affinity peptide FLPSDC*FPSV and various concentrations of the lower affinity competing peptide NLVPMVATV (0–20 µM) in presence or absence of 0.25 µM TAPBPR or TAPBPR$^{TN5}$. One of two experiments is shown. (E–G) Comparison of the dissociation of seven peptides from HLA-A*02:01fos in the presence or absence of TAPBPR or tapasin-Jun as performed in *Figure 2—figure supplement 1*. The results of three (E) and four (F,G) independent experiments were combined and are shown. Data from each experiment was processed in GraphPad Prism using one-phase exponential decay non-linear regression. The half-lives that were calculated for the dissociation of the indicated peptide in the presence of either (E) Competitor only, (F) Competitor and Tapasin-Jun or (G) Competitor and TAPBPR were plotted as a percentage of the half-life calculated for the dissociation of KLWEAESK*L in the equivalent condition. Error bars show the standard deviation of the relative half-lives measured in the different experiments. While there were slight modifications of experimental conditions between replicate experiments, results consistent with the presented results were observed.

The following figure supplement is available for figure 2:

**Figure supplement 1.** Dissociation of seven different peptides from HLA-A*02:01fos in the presence or absence of TAPBPR or Tapasin-Jun.

effect on peptide competition (*Figure 2D*). These results suggest that, like tapasin (*Chen and Bouvier, 2007*), TAPBPR can enhance the selection of high affinity peptides for binding to MHC I in vitro.

## Comparison of TAPBPR and tapasin-jun mediated peptide dissociation from HLA-A2 in vitro

Next we compared the ability of TAPBPR or tapasin-jun to enhance dissociation of FLPSDC*FPSV and a panel of six additional peptides bound to HLA-A2. The dissociation rates of all peptides were enhanced in the presence of TAPBPR and in the presence of tapasin-jun as compared to competing peptide alone (*Figure 2—figure supplement 1*). However, peptide specific differences were apparent in the ability of tapasin-jun and TAPBPR to enhance dissociation. As different concentrations of active tapasin-jun and TAPBPR proteins were present in these assays, it was not possible to directly compare the dissociation half-lives of specific peptides in the presence of tapasin-jun with TAPBPR. Therefore we compared the hierarchies of dissociation relative to one peptide (KLWEAESK*L) over four independent experiments. This revealed reproducible differences in the hierarchies of peptide dissociation between tapasin-jun and TAPBPR on HLA-A2 (*Figure 2E–G*). Interestingly the half-life hierarchy for tapasin was similar to the hierarchy observed in the absence of either cofactor (competitor only) (*Figure 2E,F*), with the exception of the dissociation of GLDDIKDLK*V whose dissociation was relatively much slower in the presence of tapasin-jun compared to the other peptides than was observed in the presence of competitor only. This suggests that tapasin enhancement of peptide dissociation generally parallels peptide-MHC complex stability, as previously reported (*Howarth et al., 2004*). However, the half-life of peptides in the presence of TAPBPR does not follow this trend (*Figure 2G*). Specifically we observed that in comparison to the other peptides, TAPBPR enhanced dissociation of KLVK*EVIAV to a much smaller extent than was apparent in the presence of tapasin-jun or absence of either cofactor (competitor only) (*Figure 2E–G*). The same was also true of FLPSDC*FPSV (*Figure 2E–G*). Taken together our results suggest firstly that both tapasin and TAPBPR enhance peptide dissociation, and secondly that there are subtle differences in regard to their peptide specificity.

## TAPBPR interacts with HLA-B as well as HLA-A molecules

We previously reported that TAPBPR has preference for HLA-A68 over HLA-B15 expressed in HeLa cells (*Boyle et al., 2013*). We wondered if TAPBPR generally exhibited preference for HLA-A allomorphs or whether TAPBPR might interact with HLA-B allomorphs. To explore this, we expressed a small panel of individual HLA allomorphs in the MHC class I negative cell line 721.221, and determined their interaction with endogenous TAPBPR, in a system in which competition between MHC class I allomorphs is absent. Immunoprecipitation of TAPBPR, followed by western blotting for MHC class I, confirmed the strong association between TAPBPR and the HLA-A allomorphs HLA-A2 and HLA-A68 (*Figure 3*). However, an association between TAPBPR and HLA-B allomorphs was also detected, although this was weaker (HLA-B8 and -B15), or below the limits of detection (HLA-B40), compared to that for HLA-A2 (*Figure 3*). Therefore, although there is clearly a strong association with those HLA-A allomorphs tested, TAPBPR interacts with a broader range of HLA molecules than originally appreciated (*Boyle et al., 2013*).

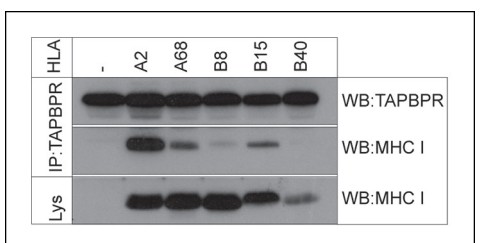

**Figure 3.** TAPBPR associates with HLA-A and HLA-B molecules. TAPBPR was isolated by immunoprecipitation (using R014) from the MHC class I negative cell line 721.221 and 721.221 stably transduced with HLA-A2, A68, B8, B15 or B40. Western blot analysis was performed for TAPBPR and the MHC class I heavy chain (using HCA2 and HC10) on lysates (labelled lys) and TAPBPR immunoprecipitates as indicated. The data is representative of three independent experiments.

## TAPBPR function on HLA-B8 in vitro

Given the interactions detected between TAPBPR and the studied HLA-B allomorphs, we asked whether TAPBPR was involved in peptide selection on HLA-B*08:01, an allomorph which interacts weakly with TAPBPR (*Figure 3*) and which is well-characterised in regard to tapasin function

(*Greenwood et al., 1994*; *Lehner et al., 1998*; *Peh et al., 1998*; *Chen and Bouvier, 2007*; *Wearsch and Cresswell, 2007*). We first asked whether TAPBPR enhanced peptide dissociation from HLA-B*08:01fos molecules. We found that dissociation of EIYK*RWIIL was enhanced in the presence of TAPBPR (*Figure 4A* and *Figure 4—figure supplement 1A*). No enhancement of the dissociation of EIYK*RWIIL was observed in the presence of TAPBPR using HLA-B8:01-T134K in which the T134 residue had been mutated to lysine, which abrogates binding of TAPBPR to MHC I (*Hermann et al., 2013*) (*Figure 4B* and *Figure 4—figure supplement 1B*), demonstrating that the enhanced peptide dissociation was dependent on a direct interaction between TAPBPR and HLA-B8. We also tested the ability of TAPBPR to enhance dissociation of two additional peptides, ELRSRK*WAI and FLRGRK*YGL from HLA-B*08:01. The dissociation of ELRSRK*WAI was not altered in the presence of TAPBPR (*Figure 4C* and *Figure 4—figure supplement 1C*) and only a very slight change in dissociation of FLRGRK*YGL was observed in the presence of TAPBPR (*Figure 4D* and *Figure 4—figure supplement 1D*). Therefore, our results suggest that TAPBPR can enhance peptide dissociation from HLA-B8 but that this effect is peptide specific. Tapasin has also been reported to exhibit peptide specificity in regard to peptide dissociation (*Chen and Bouvier, 2007*). Like TAPBPR, tapasin-jun accelerated dissociation of fluorescein isothiocyanate (FITC) labelled versions of both EIYK*RWIIL and FLRGRK*YGL from HLA-B*08:01fos, while the dissociation of ELRSRK*WAI was insensitive to tapasin-jun (*Chen and Bouvier, 2007*).

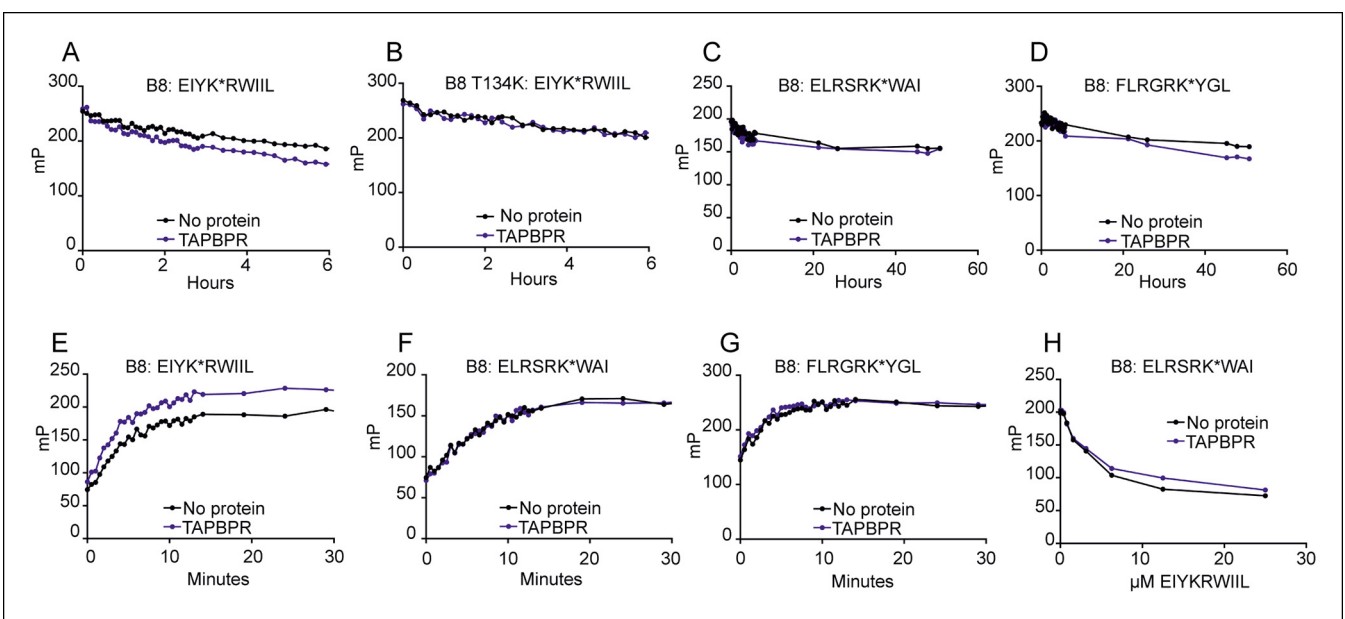

**Figure 4.** TAPBPR can function as a peptide loading catalyst and peptide editor for HLA-B8. (A–D) Dissociation, (E–G) association and (H) peptide competition of fluorescent peptides on HLA-B*08:01 in the absence or presence of TAPBPR. 2 µM HLA-B*08:01fos or HLA-B*08:01fos T134K molecules were mixed with 20 µM human β2m and made peptide receptive then loaded with 1 µM (A,B) EIYK*RWIIL (C) ELRSRK*WAI or (D) FLRGRK*YGL. Dissociation was subsequently followed after the addition of a 250 molar excess of (A,B) EIYKRWIIL (C) ELRSRKWAI, or (D) FLRGRKYGL in the absence or presence of 0.75 µM TAPBPR. A total of eight dissociation experiments have been conducted for wild-type HLA B*08:01fos, and the T134K mutant was included in six of these experiments. (E–G) 0.6 µM HLA B*08:01fos molecules were mixed with 6 µM human β2m and made peptide-receptive, then the binding of 1 µM of (E) EIYK*RWIIL, (F) ELRSRK*WAI or (G) FLRGRK*YGL was followed in the absence or presence of 0.175 µM TAPBPR. One of three experiments is shown. (H) 0.6 µM HLA-B*08:01fos molecules were mixed with 0.6 µM human β2m and were made peptide-receptive, then incubated with 0.1 µM high affinity peptide ELRSRK*WAI and various concentrations of the lower affinity competing peptide EIYKRWIIL (0–25 µM) in presence or absence of 0.3 µM TAPBPR. One of six experiments is shown. Fluorescence polarisations measurements were taken after being left at room temperature overnight. While there were slight modifications of experimental conditions between replicate experiments, results consistent with the presented results were observed.

The following figure supplement is available for figure 4:

**Figure supplement 1.** Dissociation of three different peptides from HLA B*08:01fos in the presence or absence of TAPBPR.

As with HLA-A2, we observed that TAPBPR could enhance peptide binding to peptide-receptive HLA-B*08:01fos molecules. However, this was only observed with EIYK*RWIIL (*Figure 4E*) and the binding of ELRSRK*WAI or FLRGRK*YGL to peptide receptive HLA-B*08:01fos was not altered in the presence of TAPBPR (*Figure 4F,G*). This contrasts with the results previously published by Chen and Bouvier, who observed that tapasin-jun enhanced binding of FITC labelled versions of ELRSRK*-WAI and FLRGRK*YGL to peptide-receptive HLA-B*08:01fos molecules (*Chen and Bouvier, 2007*). Taken together the data indicate that TAPBPR can enhance the loading of peptide onto HLA-B*08:01, and that this is peptide specific. Furthermore, there are differences in peptide specificity between tapasin and TAPBPR with respect to HLA-B*08:01 peptide-loading.

Finally we showed that, like tapasin (*Chen and Bouvier, 2007*), TAPBPR increased peptide exchange on HLA-B*08:01 since the low affinity peptide EIYKRWIIL became a poorer competitor against the high affinity labelled peptide ELRSRK*WAI in the presence of TAPBPR (*Figure 4H*).

## TAPBPR expression alters the peptide repertoire presented by MHC class I on cells

Given that TAPBPR functions as a peptide exchange catalyst for MHC class I in vitro, we next determined if TAPBPR expression had any effect on the peptide repertoire expressed by MHC class I molecules in cells. To do this, we created a HeLa cell line in which TAPBPR was knocked out using the clustered regularly interspaced short palindromic repeats (CRISPR) system. We found that upon treatment with IFN-γ, TAPBPR expression was induced in HeLa, but not in the HeLa-TAPBPR KO line (*Figure 5A*). Tapasin expression was not affected by the sgRNA targeted to TAPBPR (*Figure 5A*). In the absence of TAPBPR, cell surface expression of MHC class I in IFN-γ treated HeLa did not significantly change as determined by W6/32 staining and flow cytometry (*Figure 5B*). When the amino acid sequences of the peptides eluted from MHC class I molecules from IFN-γ treated HeLa and HeLa-TAPBPR KO cells were compared, the most noticeable difference was a change in the diversity of peptides isolated (*Figure 5C*). In IFN-γ treated HeLa cells, we isolated 1607 different peptides on MHC class I, compared with 2074 in IFN-γ treated HeLa-TAPBPR KO cells (*Figure 5C*). 1398 peptides were shared between the two cell lines, but 209 peptides were unique to HeLa and 676 were unique to HeLa-TAPBPR KO (*Figure 5C*). When the peptides were subdivided into those predicted to bind to HLA-A*68:02 or HLA-B*15:03 based on anchor sequences, an increased repertoire was observed for both allomorphs in the absence of TAPBPR (*Figure 5C*). Consistent with these findings, a similar pattern of increased peptide diversity was also observed in IFN-γ treated HeLa-S cells upon TAPBPR depletion using shRNA to TAPBPR (*Figure 5—figure supplement 1*).

To complement these findings we compared the amino acid sequences of peptides eluted from MHC class I molecules isolated from HeLa and HeLa cells over-expressing TAPBPR by mass spectrometry. Again the most obvious difference was a change in the diversity of peptides isolated (*Figure 5D*). We found the total number of different peptides isolated from MHC class I was decreased in cells over-expressing TAPBPR, from 819 MHC class I restricted peptides from HeLa cells down to 296 MHC class I restricted peptides isolated from HeLa cells over-expressing TAPBPR. 210 peptides were shared between the two cell lines (*Figure 5D*). Again, when these peptides were further sub-divided into those predicted to bind to HLA-A*68:02 and -B*15:03 based on their motif, an increased repertoire was observed for both allomorphs in the absence of TAPBPR (*Figure 5D*). Taken together, these results suggest that TAPBPR restricts the peptide repertoire presented on MHC class I molecules. We compared the anchor residues found at the P2 and C-terminal (PΩ) position of the shared peptides, eluted from either TAPBPR negative or TAPBPR expressing cells, which will presumably be permitted release in TAPBPR expressing cells, with unique peptides exclusively eluted from TAPBPR negative cells (HeLa or HeLa-TAPBPR KO+IFN-γ), which are therefore usually removed or restricted by TAPBPR. We found there was an enhancement of canonical anchor residues in the peptides eluted from TAPBPR expressing cells (*Figure 5E–H*). For example, in IFN-γ treated HeLa cells 53% of the shared peptides predicted to be bound to HLA-A*A68:02 (which had presumably been subjected to TAPBPR function) had classic anchor residues (P2 = threonine or valine, PΩ = valine or leucine) compared to 43% of the peptides found exclusively in TAPBPR deficient cells which are the peptides that TAPBPR usually restricts (*Figure 5E*). More striking was the restriction of peptides predicted to be bound by HLA-B*15:03 in cells over-expressing TAPBPR (*Figure 5H*) with only 5% of the peptides eluted exclusively from TAPBPR deficient cells (i.e. the TAPBPR restricted peptides) containing classical anchor motifs (P2 = glutamine or lysine, PΩ = tyrosine or phenylalanine) compared to 24% of the peptides

eluted from TAPBPR over expressing cells (i.e. the TAPBPR permitted peptides). This suggests TAPBPR restriction in vivo helps to remove some peptides of lower affinity and therefore assists in

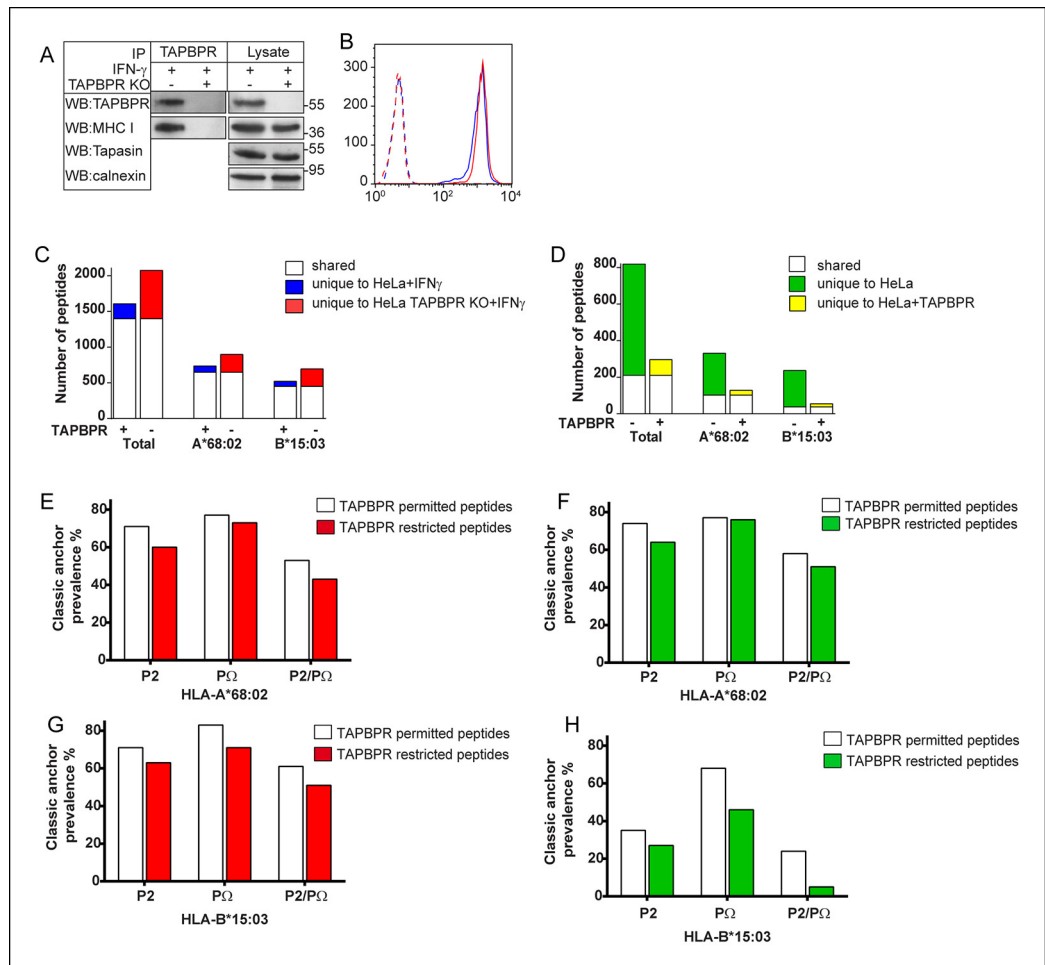

**Figure 5.** TAPBPR expression alters the peptide repertoire presented by MHC class I on cells. (A) The TAPBPR:MHC class I complex was immunoprecipitated from IFN-γ treated HeLa and HeLa-TAPBPR KO cells. Western blot analysis was performed for TAPBPR (mouse anti-TAPBPR), tapasin (Rgp48N), MHC class I (HC10) and calnexin on lysates and TAPBPR immunoprecipitates as indicated. The data is representative of three independent experiments. (B) Cytofluorometric analysis of MHC class I detected with W6/32 on IFN-γ treated HeLa (Blue line) and HeLa-TAPBPR KO cells (Red line). Staining with an isotype control on both cell lines (dashed lines) is included as a control. (C,D) Peptide-MHC class I complexes were isolated by affinity chromatography using W6/32 from (C) IFN-γ treated HeLa and HeLa-TAPBPR KO cells or (D) HeLa and HeLa overexpressing WT-TAPBPR. Eluted peptides were analysed using LC-MS/MS. Graphs show the total number of peptides and also the number of peptides assigned as HLA-A*68:02 and HLA-B*15:03 binders based on their peptide motifs using the online programme NetMHC. The number of peptides shared between the cell lines (white bar) and the number of peptides unique to each cell line (coloured bars) is shown. The data was generated from tandem MS analysis performed five times on one immunoprecipitate. Two independent biological repeats have been performed in two different cell lines (HeLa-S cells shown in *Figure 5—figure supplement 1*, and KBM-7 cells shown in *Figure 6*) in which a similar pattern of increased peptide diversity in TAPBPR depleted cells was observed. (E–H) Conservation of P2 and C-terminal anchor residues (PΩ). Plots show prevalence of classic peptide anchors on (E,F) HLA-A*68:02 or (G, H) HLA-B*15:03. Bars show classic anchor conservation at P2, PΩ and P2/PΩ combined for (E,G) shared peptides found in both IFN-γ treated HeLa and HeLa TAPBPR KO cells (white bar) (presumably permitted expression in the presence of TAPBPR) and peptides unique to HeLa TAPBPR KO + IFNγ cells (red bar) (presumably restricted in the presence of TAPBPR), (F,H) shared peptides found in both HeLa and HeLa over-expressing TAPBPR (white bar) (presumably permitted expression in the presence of TAPBPR) and peptides unique to HeLa (green bar) (presumably restricted in the presence of TAPBPR).

The following figure supplements are available for figure 5:

**Figure supplement 1.** Increased peptide diversity on MHC class I in the absence of TAPBPR in IFN-γ treated HeLa-S cells.

**Figure supplement 2.** TAPBPR expression does not influence peptide length.

improving peptide selection on MHC class I. We found TAPBPR expression did not have a dramatic effect on peptide length (*Figure 5—figure supplement 2*).

## Within a cellular environment tapasin and TAPBPR have distinct effects on peptide selection

We also compared the effect of TAPBPR depletion with tapasin deficiency on the MHC class I peptide repertoire from IFN-γ treated KBM-7 cells (see [*Hermann et al., 2013*] for knockdown efficiency). Surface expression of HLA-A2 was similar in IFN-γ treated wild-type (WT), TAPBPR depleted and tapasin deficient KBM-7 cells (*Figure 6A*). While TAPBPR depletion resulted in increased peptide diversity on HLA-A2 (*Figure 6B*), tapasin deficiency did not significantly change the total number of peptides presented on HLA-A2 (*Figure 6B*), although the peptide repertoire was significantly different between IFN-γ treated WT and tapasin deficient KBM-7 cells (*Figure 6C*). Surface expression of HLA-B40 was similar in IFN-γ treated WT and TAPBPR depleted cells, but was reduced in tapasin deficient KBM-7 cells (*Figure 6D*). TAPBPR depletion and tapasin deficiency produced different effects on peptides presented by HLA-B40 and -C03 in IFN-γ treated KBM-7 cells, with TAPBPR depletion increasing peptide diversity and tapasin deficiency decreasing the number of peptides presented (*Figure 6E,F*). A similar effect of tapasin deficiency decreasing peptide number has recently been reported for the mouse MHC I molecule H-2D$^b$ and K$^b$ (*Kanaseki et al., 2013*). Therefore, although in vitro TAPBPR

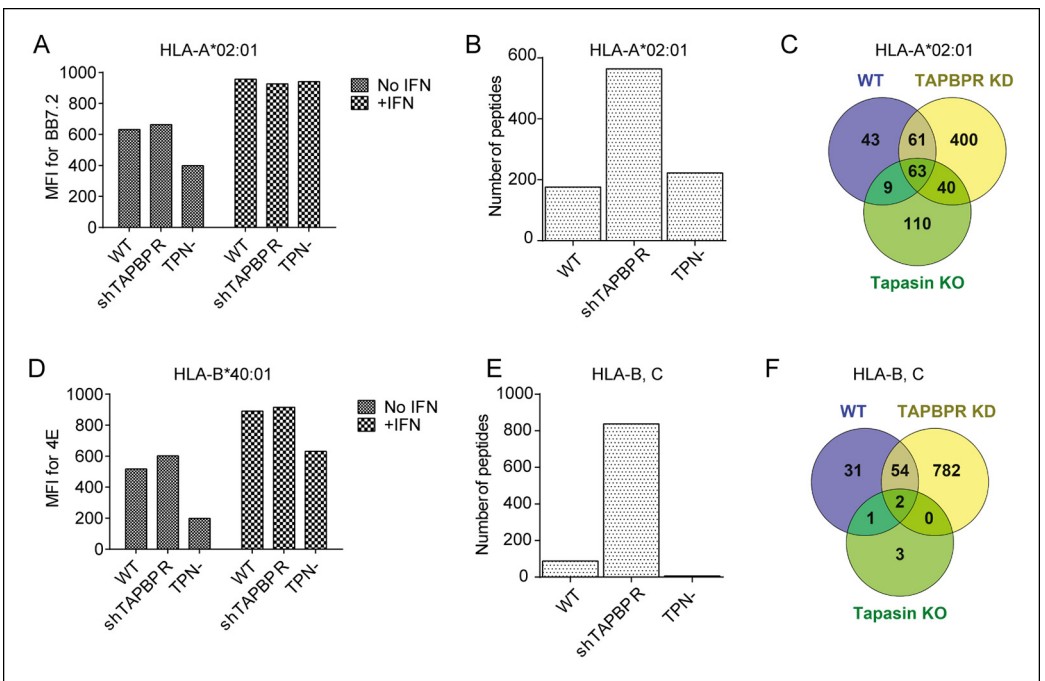

**Figure 6.** Distinct alterations to the peptide repertoire expressed on MHC class I molecules in IFN-γ treated KBM-7 cells upon TAPBPR depletion or tapasin deficiency. Cytofluorometric analysis of (**A**) HLA-A2 expression (detected with BB7.2) and (**D**) HLA-B expression (detected with 4E) on the cell surface of WT, TAPBPR stably depleted (shTAPBPR) and tapasin deficient (TPN-) KBM-7 cells treated with and without IFN-γ for 48 hr. The data is representative of three independent experiments. (**B,E**) Peptide-MHC class I complexes were isolated using affinity chromatography using (**B**) BB7.2 (HLA-A2) and (**E**) B1.23.2 (HLA-B,-C) from IFN-γ treated WT, TAPBPR stably depleted (shTAPBPR) and tapasin deficient (TPN-) KBM-7 cells. Eluted peptide were analysed using LC-MS/MS. Graphs show the total number of peptides. The data was generated from tandem MS analysis performed five times on one immunoprecipitate. Two independent biological repeats comparing the peptide repertoire expressed on MHC class I under conditions of TAPBPR competency and TAPBPR deficiency in two other cell lines have been performed (HeLa-M cells shown in *Figure 5* and HeLa-S cell shown in *Figure 5—figure supplement 1*) in which a similar pattern of increased peptide diversity upon TAPBPR depletion was observed. (**C,F**) Venn diagrams show the overlap of peptides on (**C**) HLA-A2 and (**F**) HLA-B,-C eluted from IFN-γ treated WT, TAPBPR stably depleted (TAPBPR KD) and tapasin deficient (Tapasin KO) KBM-7 cells.

and tapasin both function as peptide exchange catalysts, within the cellular environment the two molecules shape the peptide repertoire in distinct ways.

## The MHC I bound to TAPBPR is peptide-receptive

Given our finding that TAPBPR functions as a peptide exchange catalyst in vitro and restricts the peptide repertoire presented on MHC class I on cells, we next asked if the MHC class I molecules bound to TAPBPR in cells were peptide-receptive and whether they could be released from TAPBPR by high-affinity peptide. TAPBPR was isolated in complex with MHC I from IFN-γ induced KBM-7 cells (which are HLA-A2+), then incubated in the presence or absence of 100 μM of the HLA-A2 binding peptide NLVPMVATV. Western blot analysis of the proteins co-immuno-precipitated with TAPBPR revealed a >90% reduction in the amount of HLA-A2 bound to TAPBPR following the addition of the NLVPMVA-TV peptide (*Figure 7*). To ensure this was a consequence of direct peptide binding to HLA-A2, we tested NLVPMVATV peptide variants in which the anchor residues at position 2 and 9 were substituted, thus lowering the avidity of the pep-

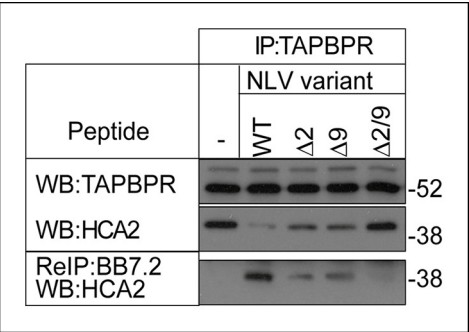

**Figure 7.** The MHC I bound to TAPBPR is peptide receptive. The TAPBPR:MHC I complex was immunoprecipitated from IFN-γ induced KBM-7 cells. Equal aliquots were divided, then incubated -/+ 100 μM (20 μg) of the indicated peptide (WT:NLVPMVATV, Δ2: NAVPMVATV, Δ9: NLVPMVATM, Δ2/9: NAVPMVATM) for 30 min at 4°C. Subsequently all eluates (- or + peptide) were re-immunoprecipitated with BB7.2. Extensive washing was performed to remove any released MHC I before denaturation. Western blot analysis was performed for TAPBPR, and HLA-A2 (using HCA2) under reducing conditions. Data shown are representative of three independent experiments.

tide. Less dissociation of HLA-A2 from TAPBPR was observed using NAVPMVATV (Δ2) and NLVPMVATM (Δ9) peptides, in which one anchor residue had been substituted, while the NAVPM-VATM (Δ2/9) peptide, in which both anchor residues were substituted was unable to induce any HLA-A2 dissociation (*Figure 7*). Next we asked whether peptide loaded HLA-A2 molecules were released from TAPBPR following peptide addition by performing a second immunoprecipitation using BB7.2 from the peptide-treated eluates. Peptide loaded HLA-A2 was released from TAPBPR upon incubation with the NLVPMVATV peptide but not in the absence of the peptide (lanes 2 and 1 respectively *Figure 7*). Only trace amounts of peptide loaded HLA-A2:β2m molecules were released from TAPBPR using the Δ2 and Δ9 peptides (lane 3 and 4, *Figure 7*), while no peptide loaded HLA-A2 was released from TAPBPR using the Δ2/9 peptide (lane 5 *Figure 7*). Together these results demonstrate that MHC I molecules bound to TAPBPR are in a peptide-receptive state and that the association between TAPBPR and MHC I is reduced by high affinity peptide resulting in peptide loaded MHC I being released from TAPBPR.

## Tapasin and TAPBPR work together to optimise surface expression of MHC class I molecules

HLA allomorphs differ in their dependency on tapasin for efficient surface expression. Previous studies have reported that, although the quantity of naturally processed peptides stably bound by HLA-A*02:01 is significantly reduced in the absence of tapasin (*Barber et al., 2001*), surface expression of HLA-A2 is relatively unaffected in the absence of tapasin (*Greenwood et al., 1994*; *Barber et al., 2001*). In contrast, HLA-B7 expression is reduced in the absence of tapasin (*Rizvi et al., 2014*). As TAPBPR is also involved in peptide selection on MHC class I molecules, we sought to examine the effect of TAPBPR depletion on surface expression of both HLA-A2 and HLA-B7 in the presence and the absence of tapasin. To investigate this, 721.221 (tapasin positive, MHC class I negative) (*Shimizu et al., 1988*) and 721.220 (tapasin negative, HLA-A, -B negative, but HLA-Cw1 positive) (*Greenwood et al., 1994*) cells were transduced with HLA-A2 and -B7 and then depleted of TAPBPR using shRNA. This resulted in a significant, although not complete, reduction of TAPBPR expression (*Figure 8A*). Tapasin expression in 721.221 cells was not altered in cells expressing shTAPBPR

(*Figure 8A*). Consistent with our previous findings in tapasin deficient KBM-7 cells (*Hermann et al., 2013*), we found an increased interaction between HLA-A2 and TAPBPR in the absence of tapasin in the 721 cell line series (compare TAPBPR IP lanes 1 and 3, *Figure 8A*) which was apparent even though the expression of TAPBPR appeared to be slightly higher in .220 cells compared to .221 cells. This phenomenon does not appear to be restricted to HLA-A2, as we similarly observed an increased interaction between HLA-B7 and TAPBPR in the absence of tapasin (compare TAPBPR IP lanes 1 and 3, *Figure 8A*). The increased association between TAPBPR and MHC class I occurs despite the fact that the steady state level of both HLA-B7 and HLA-A2 is severely reduced in the absence of tapasin (compare lysate lanes 1 and 3, *Figure 8A*). We found that the steady state level of HLA-A2 at the cell surface was not affected by TAPBPR depletion in tapasin sufficient cells (left panel *Figure 8B*). However, the depletion of TAPBPR in tapasin negative cells resulted in a ~75%

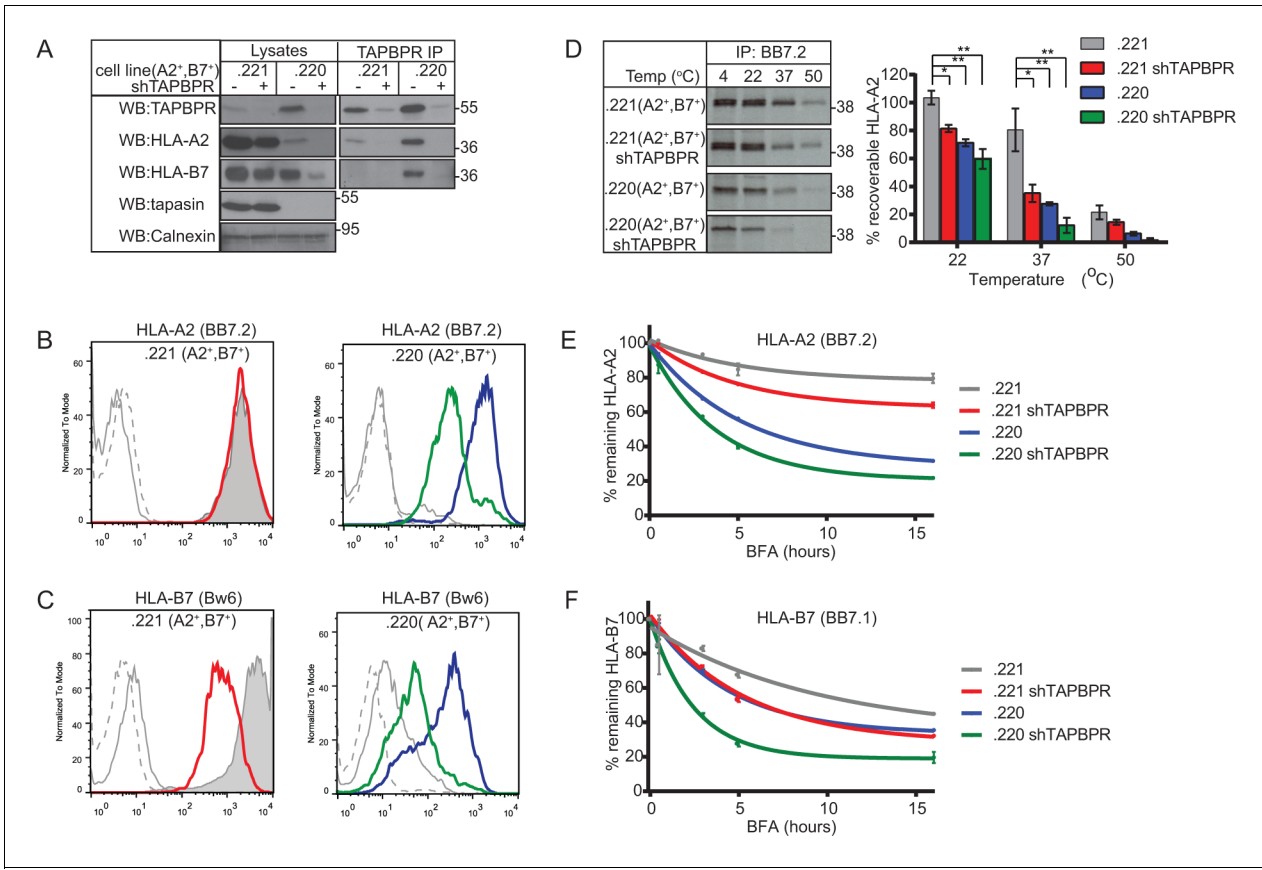

**Figure 8.** TAPBPR depletion reduces MHC class I stability. (A) The MHC class I negative cell line 721.221 and the tapasin and HLA-A and -B negative cell line 721.220 were transduced with HLA-A2 and –B7. shRNA specific for TAPBPR (shTAPBPR) was used to produce TAPBPR depleted versions of these cell lines. TAPBPR was isolated by immunoprecipitation from these four cell lines. Western blot analysis was performed for TAPBPR, tapasin, HLA-B (3B10.7), HLA-A2 (HCA2), and calnexin on lysates and TAPBPR immunoprecipitates as indicated. This data is representative of three independent repeats. Cytofluorometric analysis of (B) HLA-A2 (detected with BB7.2) or (C) HLA-B7 (detected with anti-Bw6 antibody) on .221 (A2$^+$, B7$^+$) (grey filled histogram), .221 (A2$^+$,B7$^+$) shTAPBPR (red line histogram), .220 (A2$^+$,B7$^+$) (blue line histogram), .220 (A2$^+$,B7$^+$) shTAPBPR (green line histogram). Staining on the non-transduced 721.221 and 721.220 cells with BB7.2 and Bw6 (grey solid line) or with an isotype control (grey dashed line) are included as controls. The data is representative of three independent experiments. (D) Thermal stability of HLA-A2 expressed in .221 (A2$^+$,B7$^+$) and .220 (A2$^+$, B7$^+$) -/+ stable depletion of TAPBPR (shTAPBPR). Cells were radiolabelled for 60 min with [$^{35}$S] cysteine/methionine, lysed, then equal aliquots of cleared lysates were either kept at 4°C or heated at 22°C, 37°C or 50°C for 12 min. Peptide loaded HLA-A2 was immunoprecipitated using BB7.2 post-preclear. After separation by SDS-PAGE, the signal intensity of the radiolabelled HLA-A2 band was determined by phosphorimaging. The results are representative of four independent experiments. The graph shows the percentage of peptide loaded HLA-A2 recoverable at each temperature as a percentage of the signal intensity at 4°C. Error bars show SEM from four independent experiments. Surface expression of (E) HLA-A2 (detected with BB7.2) and (F) HLA-B7 (detected with BB7.1) on cells treated with 5 μg/ml brefeldin A (BFA), which inhibits egress of newly assembled molecules from the ER, for 0, 0.5, 3, 6 and 16 hr. The level of remaining HLA-A2 and HLA-B7 at each time point is expressed as percentage of the mean fluorescence at time 0. Error bars represent SEM of duplicate samples and the data is representative of three independent experiments.

decrease in cell surface expression of HLA-A2 (right panel *Figure 8B*). In contrast, using an antibody to the Bw6 epitope we observed a significant decrease in the surface expression of HLA-B7 upon depletion of TAPBPR, even in the presence of tapasin (left panel *Figure 8C*). This is despite the poor ability of HLA-B7 to co-immunoprecipitate with TAPBPR, as compared with HLA-A2 with TAPBPR (*Figure 8A*). Only residual expression of HLA-B7 was observed upon the depletion of TAPBPR in the tapasin negative cell 721.220 (right panel *Figure 8C*). These results suggest that HLA allomorphs differ in their dependency on TAPBPR, and that tapasin and TAPBPR work together to optimise surface expression of MHC class I molecules.

## TAPBPR depletion reduces MHC class I stability

Comparison of the anchor residues in TAPBPR-permitted versus TAPBPR-restricted peptides suggests TAPBPR enhances the selection of peptides with canonical anchor residues and consequently may help reduce the cell surface presentation of peptides with lower affinity (*Figure 5E–H*). To test directly whether HLA molecules are indeed loaded with low affinity peptides in the absence of TAPBPR, we compared the stability of MHC class I molecules in the presence and absence of TAPBPR. First, the thermal stability of HLA-A2 molecules expressed in TAPBPR-competent and TAPBPR-depleted cells in the presence or absence of tapasin was investigated by heating lysates of radiolabelled cells for a short time, followed by immunoprecipitation with BB7.2 (specific for peptide loaded HLA-A2). We found that the thermal stability of HLA-A2 was reduced in TAPBPR depleted cells (compare grey bar with the red bar in *Figure 8D*). As the thermal stability of MHC class I correlates with the affinity of its peptide cargo (*Williams et al., 2002*), these results support the hypothesis that TAPBPR assists in the selection of stable peptides on MHC class I molecules. However, the effect of TAPBPR depletion on HLA-A2 thermal stability was not as severe as that observed in the absence of tapasin (compared red and blue bars in *Figure 8B*). The thermal stability of peptide loaded HLA-A2 was further decreased when TAPBPR was depleted in tapasin negative cells (green bar in *Figure 8B*). These results are consistent with the decreased steady state levels of HLA-A2 observed in TAPBPR depleted and tapasin deficient cells as shown in *Figure 8A*.

Next, to examine whether MHC class I complexes with lower stability are released onto the cell surface in the absence of TAPBPR, we examined the decay rates of MHC class I molecules expressed at the cell surface in TAPBPR-competent and TAPBPR-depleted cells. Both HLA-A2 and –B7 exhibited faster decay rates from the cell surface in TAPBPR-depleted cells (*Figure 8E,F*), suggesting that in the absence of TAPBPR MHC class I molecules containing lower affinity peptides can escape to the cell surface. For HLA-A2, tapasin deficiency had a more severe effect than TAPBPR depletion on the rate of decay (compare the blue line with the red line in *Figure 8E*) while for HLA-B7, tapasin deficiency and TAPBPR depletion had a similar effect on the decay rate (see the overlap of the blue and red line in *Figure 8F*). For both HLA-A2 and –B7 the loss of both tapasin and TAPBPR had an accumulative effect on MHC class I stability (see green line in *Figure 8E,F*). Together, these results clearly demonstrate a role for TAPBPR in the selection of stable peptides for MHC class I molecules in vivo and support the hypothesis that TAPBPR can enhance peptide optimisation in the absence of tapasin.

## Discussion

The PLC is considered to be the major site where MHC class I molecules are loaded with high affinity peptides, with the tapasin-ERp57 conjugate described as the functional unit (*Momburg and Tan, 2002*; *Williams et al., 2002*; *Howarth et al., 2004*; *Wearsch and Cresswell, 2007*). Our experiments indicate that TAPBPR is also involved in peptide selection for MHC class I molecules. Like tapasin, TAPBPR can catalyse the dissociation of peptides from peptide-MHC I complexes, enhance the loading of peptide-receptive MHC I molecules, and discriminate between peptides based on affinity in vitro. Therefore, it is now apparent that there are two MHC class I specific peptide exchange catalysts in the antigen presentation pathway intimately involved in selecting peptide for presentation to the immune system.

We found that the luminal domains of TAPBPR were able to efficiently function in MHC class I peptide selection in vitro. This suggests that the luminal region of TAPBPR alone has sufficient affinity for MHC class I to promote peptide loading and to catalyse peptide exchange and does not require other co-factors or to be artificially tethered to MHC class I to perform this function. This is

in contrast to soluble tapasin, which others have found does not associate productively with MHC class I in vitro alone, and instead requires either conjugation to ERp57 or to be artificially tethered to MHC class I (*Chen and Bouvier, 2007*; *Wearsch and Cresswell, 2007*). We speculate the difference in affinity of tapasin and TAPBPR for MHC class I may have evolved as a consequence of the distinct cellular environment the two peptide exchange catalysts function: tapasin functioning in the context of the PLC, where other proteins collectively form a mutually supportive interaction scaffold, and TAPBPR functioning outside the confines of the PLC. Although both tapasin and TAPBPR clearly function as peptide exchange catalysts in vitro, the activities of the two proteins are not identical. Firstly, we observed differences in the ability of tapasin-jun and TAPBPR to promote dissociation of peptides from HLA-A*02:01. Secondly, we found TAPBPR was unable to promote the binding of ELRSRK*WAI or FLRGRK*YGL to peptide receptive HLA-B*08:01fos molecules, whereas Chen & Bouvier found that tapasin-jun enhanced the binding of FITC labelled variants of both these peptides (*Chen and Bouvier, 2007*). This indicates there are some qualitative differences between the activity of TAPBPR and tapasin.

Clearly, a new picture concerning how peptides are selected for presentation by MHC class I molecules is emerging. Our findings are consistent with tapasin and TAPBPR working together in a sequential manner (*Figure 9*). The initial peptide loading of MHC I molecules is likely to occur within the PLC and to be assisted by tapasin. Within this relatively peptide-rich environment, peptide loading and exchange mediated by tapasin increases the affinity of peptide bound to MHC class I. We propose that outside of the PLC, the stability of peptide:MHC I complexes is monitored by TAPBPR. Those molecules loaded with high affinity peptides will either circumvent TAPBPR-mediated peptide editing completely or pass through this quality control checkpoint. For MHC class I molecules loaded with peptides of intermediate or low affinity, TAPBPR-mediated peptide editing proceeds. The unique environments in which tapasin and TAPBPR have evolved to operate in are likely to shape the peptide repertoire in different ways. The relatively peptide-rich environment of the PLC may allow tapasin to promote peptide binding and produce a relatively broad peptide repertoire. However, presumably as an MHC I molecule moves away from the PLC, the gradient of optimal peptides is likely to decrease and therefore TAPBPR-mediated peptide editing events within such an environment may favour peptide dissociation rather than association. Thus we speculate that, in a peptide rich environment, acceleration of both association and dissociation (by tapasin) contributes to editing; while in a peptide dilute environment, acceleration of dissociation only (by TAPBPR) may dominate editing, consistent with the increased peptide repertoire observed in TAPBPR deficient cells. Thus, TAPBPR may restrict peptide

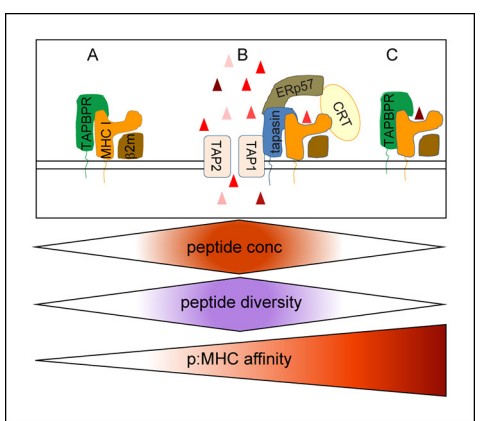

**Figure 9.** Model of the relationship between tapasin and TAPBPR in shaping the peptide repertoire expressed on MHC class I. (A) Upstream of the PLC, in an environment in which peptides suitable for MHC class I binding are likely to be at a low concentration, TAPBPR might help stabilise a peptide-receptive conformation of MHC class I, but would not directly assist in peptide loading, peptide exchange or directly increase the abundance of peptide:MHC complexes due to a shortage of peptide. (B) The optimal peptide loading environment for MHC class I is presumably within the PLC where the concentration of peptides suitable for MHC class I is the highest and where MHC I are held in a peptide-receptive conformation by tapasin. (C) Once peptide:MHC I complexes are released from the PLC, TAPBPR-mediated dissociation would attempt to remove peptide from MHC class I. If a high affinity peptide was bound to the MHC I molecules, TAPBPR would not be able to remove it. If TAPBPR can cause the dissociation of peptide, peptide exchange could ensue if suitable replacement peptides were available. However, the further away from the PLC and/or the further down the MHC class I peptide gradient TAPBPR works there may be a shortage of suitable replacement peptides. In this situation TAPBPR is unlikely to load the peptide-receptive MHC class I and may assist in the recycling of MHC I molecules. By functioning as a peptide exchange catalyst in a relatively peptide deficient environment, TAPBPR could increase the affinity of the peptide:MHC I complexes or restrict the diversity of peptide:MHC I complexes which are presented to the cell surface.

presentation by enhancing the selection of high affinity peptide on MHC class I.

Further in-depth analysis is required to dissect the intricate relationship between tapasin and TAPBPR in determining the final peptide repertoire presented on the cell surface. In particular it will be important to understand how different MHC class I allomorphs depend on these two specific chaperones. Intriguingly, we have observed differences in the functional effects of TAPBPR on peptide selection by HLA-A2 and –B7. For HLA-A2, TAPBPR catalyses both the peptide dissociation and association in vitro, in a similar manner described for the function of tapasin, albeit with different peptide preference. Therefore, there may be shared, but not identical, functionality between tapasin and TAPBPR regarding peptide selection onto HLA-A2, i.e. in a peptide abundant environment both chaperones could potentially load and edit peptides bound to HLA-A2. This may explain, at least in part, why surface expression of HLA-A2 is relatively tapasin-independent (*Lewis et al., 1998*). Surface expression of HLA-A2 is also TAPBPR-independent in the presence of tapasin, and it is not until tapasin and TAPBPR are absent or depleted that a severe effect is observed on HLA-A2 surface expression. In the absence of both tapasin and TAPBPR, surface expression of HLA-A2 resembles that of the HLA-A2 T134K mutant (*Lewis et al., 1996*; *Peace-Brewer et al., 1996*). Therefore the discovery of a peptide editing function for TAPBPR helps explain the previous discrepancies regarding the phenotype of the T134K mutant (in which the HLA-A2 mutant is unable to interact with either tapasin or TAPBPR and exhibits low cell surface expression) and tapasin-deficient cells (in which the HLA-A2 binds TAPBPR and exhibits high cell surface expression).

Our results clearly demonstrate that TAPBPR also plays a significant role in peptide selection by HLA-B molecules. This importance of TAPBPR in HLA-B biology was underappreciated previously when investigating the interaction between TAPBPR and HLA-B in HeLa cells (*Boyle et al., 2013*). Although all the HLA-B molecules investigated here (HLA-B7, -B8, -B15 and -B40) exhibit weak or transient association with TAPBPR as compared to HLA-A2 in wild-type cells, the interaction between HLA-B7 and TAPBPR was significantly increased in the absence of tapasin as shown in *Figure 8*. TAPBPR clearly influenced peptide selection on all of these HLA-B molecules: fluorescent polarisation experiments demonstrate that TAPBPR functions as a peptide loading and exchange catalyst for EIYK*RWIIL on HLA-B8; TAPBPR deficiency in IFN-γ induced HeLa and KBM-7 cells alters the peptide repertoire expressed by HLA-B15 and HLA-B40 respectively; and depletion of TAPBPR in 721.221 results in a significant decrease in the surface expression of HLA-B7, even in the presence of tapasin. Taken together, these results suggest that TAPBPR is responsible for restricting peptides on HLA-B allomorphs. Clearly, peptide selection by MHC class I molecules is far more complex than previously anticipated. Given that a single T-cell receptor (TCR) can in principle see many different peptide-MHC class I complexes (*Sewell, 2012*; *Wooldridge et al., 2012*), the two MHC specific chaperones of the MHC class I antigen presentation pathway, may be the crucial influence in selecting immune responses.

## Materials and methods

### Expression and purification of TAPBPR proteins

The luminal domains of TAPBPR and TAPBPR$^{TN5}$ (I261K) (*Hermann et al., 2013*) were cloned into pHLsec containing a C-terminal His-tag and an N-terminal signal sequence (*Aricescu et al., 2006*). To produce secreted TAPBPR and TAPBPR$^{TN5}$, 250 µg of the TAPBPR pHLsec plasmids were mixed with 1.5 ml of 1 mg/ml PEI 25K (Sigma, St Louis, MO) in 25 ml phosphate-buffered saline (PBS), followed by incubation at room temperature for 20 min. The solution was added dropwise to HEK293F cell in FreeStyle$^{TM}$ 293 Expression Medium (Gibco, Thermo Fisher Scientific, UK) seeded at a density of $1 \times 10^6$ cells/ml in a volume of 250 ml in a 1 L Erlenmeyer conical tissue flask and incubated for three days at 37°C under constant shaking at 125 rpm. The cell culture supernatant was harvested, filtered and purified using Ni-NTA affinity chromatography (Invitrogen, Thermo Fisher Scientific). Beads were isolated, washed in 20 mM Tris pH 7.4, 200 mM NaCl, 20 mM Imidazole and TAPBPR was eluted with 20 mM Tris pH 7.4, 200 mM NaCl, 300 mM Imidazole. Protein-containing fractions were analysed by SDS-PAGE followed by Coomassie staining, pooled and concentrated. The concentrate was exchanged into 50 mM Tris, 150 mM NaCl pH 8.0 buffer and separated on a Superdex S200 10/300 column (GE Heathcare, UK) by size exclusion chromatography. Protein-containing

fractions were analysed by SDS-PAGE followed by Coomassie staining, pooled and further concentrated. The concentrate was snap frozen in liquid nitrogen and stored at -80°C.

## Differential scanning fluorimetry (DSF)

DSF experiments were performed in 48-well plates using 50 µL reactions consisting of 2 µg of purified TAPBPR protein and 5x SyPRO orange dye (Invitrogen molecular probes, Thermo Fisher Scientific) in PBS pH 7.4. The melt curve was performed using Bio-Rad MiniOpticon reverse transcription-polymerase chain reaction (RT-PCR) thermal cycler between 20°C and 95°C in 1°C steps with 20 s equilibration time per step. The protein melting temperature ($T_m$) was taken as the inflexion point of the sigmoidal melting curve, obtained by curve fitting using DSF scripts (*Niesen et al., 2007*) and GraphPad Prism software.

## Production of plasmids encoding MHC I and MHC I-fos proteins

pHN1+ plasmids encoding the mature human beta 2-microglobulin protein (β2m hereafter) or the ER luminal domains of HLA A*02:01 protein with C-terminal BirA motif were obtained from Prof P Moss (University of Birmingham, UK). pGM-T7 plasmids encoding the ER luminal domains of wild-type HLA B*08:01 or the T134K point mutant, each with C-terminal fos leucine zipper sequences, were provided by Prof M Bouvier (*Chen and Bouvier, 2007*). The sequence encoding HLA-B*08:01fos or the T134K point mutant were excised by restriction enzyme digestion and sub-cloned into pET22b plasmid (Novagen, Merck Millipore, Germany). DNA encoding the ER luminal domains of HLA A*02:01 with C-terminal Fos leucine zipper sequence was created by PCR: nucleotides encoding the ER luminal domains of A*02:01 were amplified using primers 5'-GATATACCATGGGC-TCTCACTCC-3' and 5'-CGGAACCTCCCTCCCATC-3' and pHN1 A*02:01 DNA; while nucleotides encoding the Fos leucine zipper were amplified from pET22b HLA B*08:01fos using primers 5'- GG-AGGTTCCGGCGGTC-3' and 5'-CGCAAGCTTTTAATGGGCGG-3'. The purified products from both PCR reactions were used in a third PCR reaction to create A*02:01fos using primers 5'-GATATACC-ATGGGCTCTCACTCC-3' and 5'-CGCAAGCTTTTAATGGGCGG-3'.

Following agarose gel electrophoresis and digestion of the purified product with restriction enzymes the sequence encoding HLA-A*02:01fos was cloned into pET22b.

## Production of tapasin-jun proteins

DNA encoding His tagged human tapasin-jun (*Chen and Bouvier, 2007*) was amplified by PCR using 5'-GTCAGATCTGGACCCGCGGTGATCG-3' and 5'-CTCGGTACCCTAATGGTGATGGTGATG-3' primers, and ligated to pMT/BiP plasmid (Invitrogen, Thermo Fisher Scientific) following agarose gel electrophoresis and digestion of the purified product with *BglII* and *KpnI* restriction enzymes. The His6 tag present in the Jun leucine zipper portion was then replaced by PCR mutagenesis with an HA epitope using 5'-GTCAGATCTGGACCCGCGGTGATCG-3' and 5'-GTAGGTACCCTAAGCGTAG-TCTGGGACGTCGTATGGGTAGTTCATGACTTTCTG-3' primers. The resulting DNA (encoding the luminal domains of human tapasin, GGSGG linker, thrombin cleavage site, Jun leucine peptide, HA tag, and stop codon) was transferred by restriction enzyme digestion to pMT/BiP plasmid modified to encode puromycin resistance. Stable polyclonal transfectants of S2 cells were obtained by transfecting 1 µg of tapasin-jun DNA using Fugene 6 (Roche Applied Science, UK), and puromycin selection. Transfectants were adapted to EX-CELL 420 Serum-Free Medium (Sigma), and tapasin-jun expression was induced with 500 µM CuSO4. Supernatants were harvested 6 days later. Tapasin-jun was captured using anti-HA-agarose (Sigma), washed with 20 mM Tris pH7.4, 150 mM NaCl and eluted using 1 mg/ml HA peptide in 20 mM Tris pH 7.4, 150 mM NaCl. SDS PAGE electrophoresis and Coomassie staining was used to select fractions containing high concentrations of tapasin-jun, which were dialysed against 20 mM Tris-HCl pH 7.5, 150 mM NaCl at 4°C. The protein was snap frozen in liquid nitrogen and stored at -80°C.

## Peptides

The following HLA-A*02:01 binding peptides were used: the UV-labile peptide KILGFVFjV (j represents 3-amino-3-(2-nitro) phenyl-propionic acid), the fluorescent peptides FLPSDC*FPSV, KLWEAESK*L, FLLAEDTK*V, KLVK*EVIAV, YLVAEK*VTV, GLDDIKDLK*V, YLENGK*ETL (C* and K* denotes TAMRA labelled cysteine or lysine), non-labelled peptides FLPSDCFPSV, NLVPMVATV

and three variants of this peptide NAVPMVATV (Δ2), NLVPMVATM (Δ9), NAVPMVATM (Δ2/9). The following HLA-B*08:01 binding peptides were used: the UV-labile peptide FLRGRAjGL, the fluorescent peptides ELRSRK*WAI, EIYK*RWIIL or FLRGRK*YGL (K* denotes TAMRA labelled lysine), and the non-labelled peptides ELRSRKWAI, EIYKRWIIL or FLRGRKYGL. Tamra labelled and unlabelled peptides used in fluorescence polarisation were synthesised by GL Biochem Ltd (Minhang, Shanghai). All other peptides were synthesised by Peptide Protein Research Ltd (Funtley, Fareham, UK).

## Production of peptide-loaded MHC I or MHC I-fos complexes

Peptide-loaded MHC I or MHC I-fos complexes were obtained as in (*Garboczi et al., 1992*) by refolding solubilized inclusion bodies of MHC I or MHC I-fos heavy chains with solubilized inclusion bodies of human β2m and UV-labile MHC class I specific peptides.

## Fluorescence polarization experiments

Fluorescence polarization measurements were taken using an Analyst AD (Molecular Devices) with 530 nm excitation and 580 nm emission filters and 561 nm dichroic mirror. All experiments were conducted at room temperature in duplicate and used PBS supplemented with 0.5 mg/ml bovine gamma-globulin (Sigma) and 0.5 mM dithiothreitol, in a volume of 60 μl. Binding of TAMRA-labelled peptide is reported in millipolarisation units (mP) and is obtained from the equation, $mP = 1000 \times (S - G \times P)/(S + G \times P)$, where S and P are background-subtracted fluorescence count rates (S = polarised emission filter is parallel to the excitation filter; P = polarised emission filter is perpendicular to the excitation filter), and G (grating) is an instrument- and assay-dependent factor.

### Dissociation rate measurements

Peptide-receptive HLA-A*02:01 and HLA-B*08:01 were obtained by exposing the monomeric MHC class I complexes loaded with UV labile peptides to ~360 nm light for 20 min at 4°C ('UV exposed' hereafter). MHC class I molecules were allowed to bind to fluorescent peptides overnight at 4°C. Dissociation of the fluorescent peptide was subsequently followed at room temperature after the addition of excess non-labelled competitor peptide in the absence or presence of TAPBPR, TAPBPR[TN5] or tapasin-jun.

### Association rate measurements

The binding of fluorescent peptides to UV-exposed MHC class I was monitored in the absence and the presence of TAPBPR or TAPBPR[TN5].

### Peptide competition experiments

The binding of fluorescent peptide to MHC class I was monitored in the presence of a titration of non-labelled competitor peptides without and with TAPBPR or TAPBPR[TN5]. Fluorescence polarizations measurements were taken after being left at room temperature overnight.

## Antibodies

The following TAPBPR specific antibodies were used: PeTe4 (a mouse mAb raised against the luminal domains of human TAPBPR which recognise a native conformation of the protein) (*Boyle et al., 2013*), R014 (a rabbit polyclonal raised against the luminal domains of human TAPBPR), R021 (a rabbit polyclonal raised against the cytoplasmic tail of human TAPBPR), a mouse anti-TAPBPR mAb raised against the membrane distal domain of TAPBPR (ab57411, Abcam, UK). The following MHC class I specific antibodies were used: HC10 (a mouse mAb that recognises a PxxWDR motif at aa 57–62 in the α1 domain of the MHC class I heavy chain, found commonly in HLA-B and –C, and also in a few HLA-A allomorphs) (*Stam et al., 1986*; *Perosa et al., 2003*), HCA2 (a mouse mAb that recognises HLA allomorphs containing a xLxTLRGx motif at aa 77–84 on the α1 domain) (*Stam et al., 1990 Sernee et al., 1998*), 3B10.7 (a rat monoclonal antibody with broad specificity for HLA I independent of conformation) (*Lutz and Cresswell, 1987*), BB7.2 (a mouse mAb specific for β2m bound and peptide loaded HLA-A2), Bw6 (a mouse mAb specific for MHC class I allomorphs containing the Bw6 epitope SLRNLRG at aa 77–83) (One Lambda, Thermo Fisher Scientific, Canoga Park, CA), BB7.1 (a

mouse mAb specific for HLA-B7) (AbDSerotec), W6/32 (a pan MHC I mAb which recognises a conformation epitope on the MHC I α2 domain, dependent on β2m and peptide) (*Barnstable et al., 1978*), 4E (a mouse monoclonal reactive against HLA-B as well as a limited number of HLA-A molecules (A29, Aw30, Aw31, Aw32) (*Yang et al., 1984*), B1.23.2 (a mouse mAb that recognises both β2m association and -free HLA-B,-C as well as some HLA-A heavy chains) (*Rebaï and Malissen, 1983*; *Apps et al., 2009*). Other antibodies used were Pasta1 (a mAb which recognises native tapasin) (*Dick et al., 2002*) and Rpg48N (tapasin specific rabbit polyclonal) (both kind gifts from Peter Cresswell, Yale University School of Medicine, New Haven, CT), and rabbit anti-calnexin (Enzo life Sciences, UK). Primary antibodies were detected with horseradish peroxidase (HRP) conjugated secondary antibodies (Dako, UK) and goat-anti-mouse Alexa 647 (molecular probes). In addition isotype control antibodies were also used (Dako).

## Cell lines

HEK293T were maintained in Dulbecco's modified eagle's medium (DMEM;Gibco), HeLa-M, HeLa-S, HeLa-S depleted of TAPBPR using shRNA (*Boyle et al., 2013*), 721.221 and 721.220 cells were maintained in Roswell Park Memorial Institute (RMPI) 1640 media (Gibco), and KBM-7 cells were maintained in Iscove's Modified Dulbecco's Medium (Gibco) all supplemented with 10% fetal calf serum, 100 U/ml penicillin and 100 µg/ml streptomycin (Gibco) at 37°C and 5% $CO_2$. To induce the expression of endogenous TAPBPR, cells were treated with 50 U/ml of IFN-γ (Peprotech, UK) at 37°C for 48–72 hr.

## Plasmids and transductions

HLA-A*02:01, -A*68:01, -B*08:01, -B*15:10 and –B*40:02 were cloned into the lentiviral vector pHRSINcPPT-SGW. HLA-B*07:02 was cloned into the lentiviral vector pHRSIN-C56W-UbEM producing HLA-B7 under the spleen focus-forming virus promotor and the green fluorescent protein (GFP)-derivative protein emerald under an ubiquitin promotor. For TAPBPR depletion, the lentiviral shRNA plasmid V2LHS_135531 on the pGIPZ backbone was used (Open Biosystems, GE healthcare). Lentiviral plasmids were transfected into HEK-293T cells along with the packaging plasmid pCMVR8.91 and envelope plasmid pMD-G using TransIT-293 (Mirus, Madison, WI). Forty eight and seventy two hours post-transfection, the filtered supernatants were used to transduce HeLa, 721.221 or 721.220 cells. For HLA molecules transduced into 721.221, the transduction efficiency was determined by the surface expression of MHC class I using W6/32 antibody with flow cytometry. To make the 721.221 and 721.220 expressing both HLA-A2 and –B7, HLA-A2 was first stably transduced into the cell lines with transduction efficiency determined by surface expression of HLA-A2 using BB7.2 antibody with flow cytometry. These HLA-A2 positive cells were subsequently transduced with HLA-B7 pHRSIN-C56W-UbEM with transduction efficiency determined by GFP expression in the cells. To select cells depleted of TAPBPR puromycin selection was used post transduction with V2LHS_135531.

## Generation of HeLa TAPBPR knock-out cell lines using CRISPR

HeLa TAPBPR KO lines were generated using CRISPR as previously described (*Ran, et al., 2013*). To target TAPBPR expression, sgRNAs were chosen using the CRISPR design tool (http://crispr.mit.edu), which bind in exon 2 of TAPBPR (Crispr7: GCGAAGGACGGTGCGCACCG) and cloned into pSpCas9(BB)-2A-Puro. To generate HeLa TAPBPR KO cell lines, cells were seeded at a confluence of 80% in 6 well format and transfected with TAPBPR-CRISPR plasmids in the absence of serum using Lipofectamine2000 (Invitrogen, Thermo Fisher Scientific). Twenty four hours after transfection, the medium was replaced with complete DMEM containing 4 µg/ml puromycin (Invivogen, San Diego, CA). After 48 hr, the medium was replaced with complete DMEM without puromycin. Subsequently, the clones were selected and the absence of TAPBPR protein was verified by immunoprecipitation/western blot analysis.

## Immunoprecipitation and western blot analysis

Harvested cells were washed in PBS, pelleted, then lysed in either 1% digitonin (Merck Millipore) Tris-buffered saline (TBS) (20 mM Tris-HCl pH7.4, 150 mM NaCl, 5 mM $MgCl_2$, 1 mM ethylenediaminetetraacetic acid) supplemented with 10 mM N-ethylmaleimide (NEM) (Sigma), 1mM

phenylmethylsulfonyl fluoride (PMSF; Sigma) and protease inhibitor cocktail (Roche) for 30 min at 4°C. Nuclei and cell debris were pelleted by centrifugation at 13,000 × g for 10 min and supernatants were precleared on IgG-sepharose (GE Healthcare, United Kingdom) and Protein A Sepharose (Generon, UK) for 1 hr at 4°C with rotation. Immunoprecipitation was performed with the indicated antibody and Protein A Sepharose for 1 hr at 4°C with rotation. Following immunoprecipitation, the beads were washed thoroughly in 0.1% detergent-TBS to remove the unbound protein. All samples were heated at 80°C for 10 min in reducing sample buffer (125 mM Tris-HCL pH 6.8, 4% SDS, 20% glycerol, 0.04% bromophenol blue with 100 mM β-mercaptoethanol). Proteins were separated by SDS-PAGE and transferred onto an Immobilon transfer membrane (Millipore, Billerica, MA). Membranes were blocked using 5% (w/v) dried milk, 0.1% (v/v) Tween 20 in PBS for 30 min, followed by incubation with the indicated primary antibody for 1 hr. After washing, membranes were incubated with species-specific HRP conjugated secondary antibodies, before detection by enhanced chemiluminescence reagent (GE Healthcare).

## MHC class I thermal stability assays

Cells were starved in methionine and cysteine free RPMI for 30 min at 37°C, then labelled using EasyTag Express [35S]-protein labeling mix (Perkin Elmer) for 60 min at 37°C. Cells were lysed in 1% Triton X-100 (Sigma) TBS containing 10 mM NEM, 1 mM PMSF and protease inhibitors for 30 min at 4°C. Equal aliquots of clarified cell lysates were either kept at 4°C or heated at 22, 37 or 50°C for 12 min, before returning to 4°C. Immunoprecipitation and SDS-PAGE were performed as above. Gels were subsequently fixed in 12% acetic acid, 40% methanol and dried. Images were obtained using a phosphor screen (Perkin-Elmer) or on film. PhosphorImager analysis was performed using Typhoon Trio variable mode imager (GE Healthcare) together with Image-QuantTL software. Densitometry of the MHC class I HC band was performed. The amount of recoverable HLA-A2 remaining at each temperature was determined as a percentage of the signal intensity at 4°C. Graphs were generated using GraphPad Prism software. High-resolution images were obtained using film.

## Flow cytometry

Cells were washed in PBS, then incubated for 20 min at 4°C with W6/32, BB7.2, BB7.1, 4E or a HLA-Bw6 specific antibody. Isotype control antibodies were used as negative controls. Cells were washed in ice-cold PBS, then the bound primary antibody was subsequently detected with species-specific Alexa Fluor 647 secondary antibodies (Molecular Probes). Cells were analysed on a BD FACS Calibur 4-(BD Biosciences, East Rutherford, NJ) colour analyser and data were analysed using FlowJo software. For MHC class I cell surface decay experiments, 5 µg/ml brefeldin A was incubated with cells for 0, 0.5, 3, 6, and 16 hr followed by staining with BB7.2 or BB7.1.

## MHC class I peptide analysis

HLA ligands from $5 \times 10^8$ cells per cell line were isolated by immunoaffinity chromatography. Cells were lysed in buffer containing PBS, 0.6% 3-[(3-cholamidopropyl)dimethylammonio]-1-propanesulfonate (CHAPS), and Complete protease inhibitor, shaken for 1 hr at 4°C and subsequently sonicated for 1 min. Following centrifugation for 1.5 hr to remove debris, the supernatant was applied on affinity columns overnight at 4°C. Columns were previously prepared by coupling antibodies to CNBr-activated Sepharose (GE Healthcare, Buckinghamshire, England) (1 mg antibody/40 mg Sepharose). The antibody W6/32 was used to isolate all HLA class I alleles in HeLa cells. For KBM-7 cells, BB7.2 was used to isolate HLA-A2 and B1.23.2 was used to isolate HLA-B and -C molecules.

On the second day the columns were eluted in eight steps using 0.2% trifluoroacetic acid. The eluate was passed through a 10 kDa filter (Merck Millipore, Darmstadt, Germany) to yield the HLA-ligands in solution. The filtrate was desalted with C18 ZipTips (Merck Millipore, Darmstadt, Germany) and subsequently concentrated using a vacuum centrifuge (Bachofer, München, Germany). Sample volume was adjusted for measurement by adding 1% ACN/0.05% TFA (v/v). With an injection volume of 5 µl HLA ligands were loaded (100 µm × 2 cm, C18, 5 µm, 100 Å) and separated (75 µm × 50 cm, C18, 3 µm, 100 Å) on Acclaim Pepmap100 columns (Dionex, Sunnyvale, CA) using an Ultimate 3000 RLSCnano uHPLC system (Dionex). A gradient ranging

from 2.4 to 32% of ACN/H2O with 0.1% formic acid was used to elute the peptides from the columns over 140 min at a flow rate of 300 nl/min. Online electrospray ionisation (ESI) was followed by tandem mass spectrometry (MS) analysis in a LTQ Orbitrap XL instrument (Thermo-Fisher Scientific, Bremen, Germany). Survey scans were acquired in the Orbitrap mass analyzer with a resolution of 60,000 and a mass range of 400–650 m/z. Peptides with a charge state other than 2+ or 3+ were rejected from fragmentation. Fragment mass spectra of the five most intense ions of each scan cycle were recorded in the linear ion trap (top5 CID). Normalised collision energy of 35%, activation time of 30 ms and isolation width of 2 m/z was utilised for fragment mass analysis. Dynamic exclusion was set to 1 s. The rawwere processed against the human proteome as comprised in the Swiss-Prot database (www.uniprot.org, status: Dec12th, 2012; 20.225 reviewed sequences contained) using MASCOT server version 2.3.04 (Matrix Science, Boston, MA) and Proteome Discoverer 1.4 (Thermo Fisher Scientific). Oxidation of methionine was allowed as dynamic peptide modification. A mass tolerance of 5 ppm or 0.5 Da was allowed for parent- and fragment masses respectively. Filtering parameters were set to a Mascot Score<20, search engine rank = 1, peptide length of 8-12 AA, achieving a false discovery rate (FDR) of 5% as determined by an inverse decoy database search. The netMHC programme (http://www.cbs.dtu.dk/services/NetMHC/) was used to predict the binding of peptides to specific HLA allomorphs (*Lundegaard et al., 2008a*, *2008b*). To determine classic peptide anchor conservation, TAPBPR permitted peptides (shared peptides) and TAPBPR restricted peptides (peptides unique to HeLa-TAPBPR KO cells or peptides unique to HeLa cells) were compared and the frequency of classic anchors at P2 or PΩ calculated for both HLA-A*68:02 or HLA-B*15:03.

## Additional information

### Funding

| Funder | Grant reference number | Author |
|---|---|---|
| Wellcome Trust | Senior Research Fellowship. 104647 | Louise H Boyle |
| Royal Society | University Research Fellowship, UF100371 | Janet E Deane |
| Cancer Research UK | Programme Grant, C7056A | Andy van Hateren Tim Elliott |
| Wellcome Trust | Research Career Development Fellowship, 085058 | Louise H Boyle |
| Wellcome Trust | PhD studentship, 089563 | Clemens Hermann |
| Deutsche Forschungsgemeinschaft | SFB 685 | Nico Trautwein Stefan Stevanović |
| Wellcome Trust | Programme Grant, 089821 | John Trowsdale |

The funders had no role in study design, data collection and interpretation, or the decision to submit the work for publication.

### Author contributions

CH, AvanH, JED, Acquisition of data, Analysis and interpretation of data, Drafting or revising the article; NT, SS, Acquisition of data, Analysis and interpretation of data; AN, Acquisition of data, Drafting or revising the article; PJD, Assistance in protein expression and purification, Drafting or revising the article; JT, Conception and design, Drafting or revising the article; TE, Conception and design, Analysis and interpretation of data, Drafting or revising the article; LHB, Conception and design, Acquisition of data, Analysis and interpretation of data, Drafting or revising the article, Contributed unpublished essential data or reagents

### Author ORCIDs

Clemens Hermann, http://orcid.org/0000-0002-0009-9501
Andy van Hateren, http://orcid.org/0000-0002-3915-0239
Nico Trautwein, http://orcid.org/0000-0003-3292-8850

Andreas Neerincx, http://orcid.org/0000-0002-6902-5383
Patrick J Duriez, http://orcid.org/0000-0003-1814-2552
Stefan Stevanović, http://orcid.org/0000-0003-1954-7762
John Trowsdale, http://orcid.org/0000-0002-0150-5698
Janet E Deane, http://orcid.org/0000-0002-4863-0330
Tim Elliott, http://orcid.org/0000-0003-1097-0222
Louise H Boyle, http://orcid.org/0000-0002-3105-6555

## Additional files

### Major datasets

The following datasets were generated:

| Author(s) | Year | Dataset title | Dataset ID and/or URL | Database, license, and accessibility information |
|---|---|---|---|---|
| Hermann C, van Hateren A, Trautwein N, Neerincx A, Duriez PJ, Stevanović S, Trowsdale J, Deane JE, Elliott T, Boyle LH | 2015 | Data from: TAPBPR alters MHC class I peptide presentation by functioning as a peptide exchange catalyst | http://dx.doi.org/10.5061/dryad.487j9 | Available at Dryad Digital Repository under a CC0 Public Domain Dedication |

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
