## [Decision Letter]

Thank you for submitting your work entitled "TAPBPR alters MHC class I peptide presentation by functioning as a peptide exchange catalyst" for peer review at *eLife*. Your submission has been favorably evaluated by Tadatsugu Taniguchi (Senior Editor) and three reviewers, one of whom is a member of our Board of Reviewing Editors. One of the three reviewers, Nilabh Shastri, has agreed to share his identity.

The reviewers have discussed the reviews with one another and the Reviewing editor has drafted this decision to help you prepare a revised submission.

Summary:

Clemens and colleagues report new details of what is emerging as a key step in the antigen-processing pathway for generating peptide-loaded MHC class I molecules. They show that TAPBPR, a tapasin-like protein which was discovered earlier by some members of this group, is a peptide editor for selective loading of MHC I molecules. Unlike tapasin, which functions in the peptide-loading complex (PLC) in the endoplasmic reticulum (ER), TAPBPR apparently carries out its peptide editing functions outside the PLC and this editing function does not overlap with that of tapasin. This is a significant advance in our understanding of the peptide-selection mechanisms in the MHC class I antigen processing pathway.

Essential revisions:

The authors are to be congratulated on a very high-quality and interesting manuscript. Only the relatively minor revisions listed below have been requested:

1) The actual peptide sequences should be made available to the reader.

2) In the absence of TAPBPR, the number of different peptides is enhanced, suggesting that these peptides were edited out in the presence of TAPBPR. Although not explicitly stated, such peptides are likely of low affinity. This could be directly tested with some of the synthetic peptides from the list in their fluorescence peptide binding/competition assays.

3) If the peptide-HLA is indeed loaded with low affinity peptides in absence of TAPBPR, then such complexes should be less stable on the cell surface. This could be easily tested by measuring the decay of peptide-loaded HLAs from the cell surface in brefeldin A treated cells which would not export newly assembled HLA molecules. Faster decay of HLA from the cell surface would strengthen the conclusion that the new peptides that arise in TAPBPR-deficient cells are low affinity. This evidence should further support the lower thermal stability data for MHC that actually makes it to the cell surface.

4) Reproducibility: the authors need to clearly indicate how reproducible the presented data is. How many times the experiments were performed and what the error bars shown represent.

---

## [Author Response]

*The authors are to be congratulated on a very high-quality and interesting manuscript. Only the relatively minor revisions listed below have been requested:*

*1) The actual peptide sequences should be made available to the reader.*

We now include lists containing the peptide sequences from all mass spectrometry experiments so that they are available to the reader.

*2) In the absence of TAPBPR, the number of different peptides is enhanced, suggesting that these peptides were edited out in the presence of TAPBPR. Although not explicitly stated, such peptides are likely of low affinity. This could be directly tested with some of the synthetic peptides from the list in their fluorescence peptide binding/competition assays.*

Thank you for the suggestion to test some of the peptides identified by mass spectrometry in our fluorescence polarisation experiments. While it is possible to do these experiments, we feel the results would be entirely dependent on the particular peptide we happen to pick from the peptide pool. As our peptide lists have thousands of different peptides and we can only analyse perhaps ten of these peptides in the assay (which would only represent 1% of the total peptide pool) we feel this approach might not be reflective of TAPBPR function on the whole population of peptide and we might inadvertently skew the data depending on the particular 1% of peptides we selected to test in our assay.

Therefore, as an alternative approach to examine whether TAPBPR is involved in editing out peptides of low affinity, we have compared the anchor residues of peptides shared in TAPBPR-competent and TAPBPR-depleted cells, which are therefore indicative of peptides permitted release through the TAPBPR filter/restriction step, with the anchor residues of unique peptides found in TAPBPR negative cells, which are indicative of peptides which are removed or restricted by TAPBPR expression. This new data can be found Figure 5 (sections E-H). Using this approach, we observed a larger percentage of peptides which are permitted release by TAPBPR have classic anchor residues compared to the peptides restricted by TAPBPR. These results suggest there is some enhancement of peptide anchors on MHC I by TAPBPR and hence TAPBPR has the potential to remove some peptides with lower affinity. We hope this new data analysis fully answers the reviewers point regarding TAPBPR editing out some peptides of lower affinity within a cellular environment.

*3) If the peptide-HLA is indeed loaded with low affinity peptides in absence of TAPBPR, then such complexes should be less stable on the cell surface. This could be easily tested by measuring the decay of peptide-loaded HLAs from the cell surface in brefeldin A treated cells which would not export newly assembled HLA molecules. Faster decay of HLA from the cell surface would strengthen the conclusion that the new peptides that arise in TAPBPR-deficient cells are low affinity. This evidence should further support the lower thermal stability data for MHC that actually makes it to the cell surface.*

Thank you for suggesting the BFA decay experiments to test the effect of TAPBPR depletion on the stability of MHC class I at the cell surface. We have now performed these experiments comparing the stability of two MHC class I molecules, HLA-A2 and HLA-B7, in TAPBPR-competent and TAPBPR-depleted cells (and in the presence and absence of tapasin as a control) and do indeed find an increase in the rate of decay of MHC class I in the absence of TAPBPR. This new data can be found in Figure 8 (section E and F). As we observe a faster rate of decay of HLA from the cell surface in TAPBPR depleted cells, this new data strengthens the conclusion that at least some of the new peptides that arise in TAPBPR-deficient cells are low affinity.

4) Reproducibility: the authors need to clearly indicate how reproducible the presented data is. How many times the experiments were performed and what the error bars shown represent.

We have added additional text in the figure legends to indicate how many times each experiment was performed and what the error bars shown represent.